# Estimating Learnability in the Sublinear Data Regime

**Weihao Kong**
Stanford University
whkong@stanford.edu

**Gregory Valiant**
Stanford University
gvaliant@cs.stanford.edu

## Abstract

We consider the problem of estimating how well a model class is capable of fitting a distribution of labeled data. We show that it is possible to accurately estimate this "learnability" even when given an amount of data that is too small to reliably learn any accurate model. Our first result applies to the setting where the data is drawn from a $d$-dimensional distribution with isotropic covariance, and the label of each datapoint is an arbitrary noisy function of the datapoint. In this setting, we show that with $O(\sqrt{d})$ samples, one can accurately estimate the fraction of the variance of the label that can be explained via the best linear function of the data. For comparison, even if the labels are noiseless linear functions of the data, a sample size linear in the dimension, $d$, is required to *learn* any function correlated with the underlying model. Our estimation approach also applies to the setting where the data distribution has an (unknown) arbitrary covariance matrix, allowing these techniques to be applied to settings where the model class consists of a linear function applied to a nonlinear embedding of the data. In this setting we give a consistent estimator of the fraction of explainable variance that uses $o(d)$ samples. Finally, our techniques also extend to the setting of binary classification, where we obtain analogous results under the logistic model, for estimating the classification accuracy of the best linear classifier. We demonstrate the practical viability of our approaches on synthetic and real data. This ability to estimate the explanatory value of a set of features (or dataset), even in the regime in which there is too little data to realize that explanatory value, may be relevant to the scientific and industrial settings for which data collection is expensive and there are many potentially relevant feature sets that could be collected.

## 1 Introduction

Given too little labeled data to learn a model or classifier, is it possible to determine whether an accurate classifier or predictor exists? For example, consider a setting where you are given $n$ datapoints with real-valued labels drawn from some distribution of interest, $D$. Suppose you are in the regime in which $n$ is too small to learn an accurate prediction model; might it still be possible to estimate the performance that would likely be obtained if, hypothetically, you were to gather more data, say a dataset of size $n' \gg n$ and train a model on that data? We answer this question affirmatively, and show that in the settings of linear regression and binary classification via linear (or logistic) classifiers, it *is* possible to estimate the likely performance of a (hypothetical) predictor trained on a larger hypothetical dataset, even given an amount of data that is sublinear in the amount that would be required to *learn* such a predictor.

For concreteness, we begin by describing the flavor of our results in a very basic setting: learning a noisy linear function of high-dimensional data. Suppose we are given access to independent samples from a $d$-dimensional isotropic Gaussian, and each sample, $\mathbf{x} \in \mathbf{R}^d$ is labeled according to a noisy linear function $y = \langle \mathbf{x}, \beta \rangle + \eta$, where $\beta$ is the true model and the noise $\eta$ is drawn (independently) from a distribution of (unknown) variance $\delta^2$. One natural goal is to estimate the signal to noise ratio, $1 - \delta^2/\mathbf{Var}[Y]$, namely estimating how much of the variation in the label we could hope to explain.

Even in the noiseless setting ($\delta = 0$), it is information theoretically *impossible* to learn any function that has even a small constant correlation with the labels unless we are given an amount of data that is linear in the dimension, $d$. Nevertheless, as was recently shown by Dicker [6] in this Gaussian setting with independent noise, it is possible to estimate the magnitude of the noise, $\delta$, and variance of the label, given only $O(\sqrt{d})$ samples.

Our results (summarized in Section 1.1), explore this striking ability to estimate the "learnability" of a distribution over labeled data based on relatively few samples. Our results significantly extend previous results and the results of Dicker in the following senses: 1) We present a unified approach that yields accurate estimation of this learnability when $n = o(d)$ which applies even when the $\mathbf{x}$ portion of the datapoints are drawn from a distribution with arbitrary (unknown) covariance. This is *very* surprising—and was conjectured to be impossible [14]—because the best linear model can not be approximated with $o(d)$ data, nor can the covariance be consistently estimated with $o(d)$ datapoints. 2) Agnostic setting: Our techniques do not require any distributional assumptions on the label, $y$, in contrast to most previous work that assumed $y$ is a linear function plus independent noise (which is not a realistic assumption for many of the practical settings of interest). Instead, our approach directly estimates the fraction of the variance in the label that can be explained via a linear function of $\mathbf{x}$. 3) Binary classification setting: Our techniques naturally extend to the setting of binary classification, provided a strong distributional assumption is made—namely that the data is drawn according to the logistic model (see Section 1.1 for a formal description of this model).

Throughout, we focus on *linear* models and classifiers. Because some of our results apply when the covariance matrix of the distribution is non-isotropic (and non-Gaussian), the results extend to the many non-linear models that can be represented as a linear function applied to a non-linear embedding of the data, for example settings where the label is a noisy polynomial function of the features. Still, our assumptions on the data generating distribution are very specific for the binary classification results, and our algorithm for this setting does not apply if the two classes have unequal probabilities. We are optimistic that our techniques may be extended in future work to address that setting, as well as more general function classes and losses.

**Motivating Applications: Estimating the value of data and dataset selection.** In some data-analysis settings, the ultimate goal is to quantify the signal and noise—namely understand how much of the variation in the quantity of interest can be explained via some set of explanatory variables. For example, in some medical settings, the goal is to understand how much disease risk is associated with genomic factors (versus random luck, or environmental factors, etc.). In other settings, the goal is to accurately predict a quantity of interest. The key question then becomes "what data should we collect—what features or variables should we try to measure?" The traditional pipeline is to collect a lot of data, train a model, and then evaluate the value of the data based on the performance (or improvement in performance) of the model.

Our results demonstrate the possibility of evaluating the explanatory utility of additional features, even in the regime in which too few data points have been collected to leverage these data points to learn a model. For example, suppose we wish to build a predictor for whether or not someone will get a certain disease. We could begin by collecting a modest amount of genetic data (e.g. for a few hundred patients, record the presence of genetic abnormalities for each of the 20k genes), and a modest amount of epigenetic data. Even if we have data for too few patients to learn a good predictor, we can at least evaluate how much the model would improve if we were to collect a lot more genetic data, versus collecting more epigenetic data. This ability to *explore* the potential of different features with less data than would be required to *exploit* those features seems extremely relevant to the many industry and research settings where it is expensive or difficult to gather data.

Alternately, these techniques could be leveraged by data providers in the context of a "verify then buy" model: Suppose I have a large dataset of customer behaviors that I think will be useful for your goal of predicting customer clicks/purchases. Before you purchase access to my dataset, I could give you a tiny sample of the data—too little to be useful to you, but sufficient for you to verify the utility of the dataset.

One final downstream application of our techniques is to the data aggregation and "federated learning" settings. The approaches of this work can be re-purposed to measure the extent to which two or more labeled datasets have the same (or similar) labeling functions, even in the regime in which there is too little data to learn such labeling functions. (This can be accomplished, for example, by applying our techniques of this paper to the aggregate of the datasets versus individually and seeing whether the

signal to noise ratio degrades upon aggregation.) Such a primitive might have fruitful applications in realm of federated learning, since one of the key questions in such settings is how to decide which entities have similar models, and hence which subsets of entities might benefit from training a model on their combined data.

## 1.1 Summary of Results

Our first result applies to the setting where the data is drawn according to a $d$ dimensional distribution with identity covariance, and the labels are noisy linear functions. This result generalizes the results of Dicker [6] and Verzelen and Gassiat [14] beyond the Gaussian setting. Provided there are more than $O(\sqrt{d})$ datapoints, the magnitude of the noise can be accurately determined:

**Proposition 1.** *[Slight generalization of Lemma 2 in [6] and Corollary 2.2 in [14]] Suppose we are given $n$ labeled examples, $(\mathbf{x}_1, y_1), \ldots, (\mathbf{x}_n, y_n)$, with $x_i$ drawn independently from a d-dimension distribution of mean zero, identity covariance, and fourth moments bounded by $C$. Assuming that each label $y_i = \mathbf{x}_i\beta + \eta$, where the noise $\eta$ is drawn independently from an (unknown) distribution $E$ with mean 0 variance $\delta^2$, and the labels have been normalized to have unit variance. There is an estimator $\hat{\delta}^2$, that with probability $1 - \tau$, approximates $\delta^2$ with additive error $O(C\frac{\sqrt{d+n}}{\tau n})$.*

The fourth moment condition of the above theorem is formally defined as follows: for all vectors $\mathbf{u}, \mathbf{v} \in \mathbf{R}^d$, $\mathbf{E}[(\mathbf{x}^T\mathbf{u})^2(\mathbf{x}^T\mathbf{v})^2] \leq C\mathbf{E}[(\mathbf{x}^T\mathbf{u})^2]\mathbf{E}[(\mathbf{x}^T\mathbf{v})^2]$. In the case that the data distribution is an isotropic Gaussian, this fourth moment bound is satisfied with $C = 3$.

In the above setting, it is information theoretically impossible to approximate $\beta$, or accurately predict the $y_i$'s without a sample size that is *linear* in the dimension, $d$. As we show, the above result is also optimal, to constant factors, in the constant-error regime: no algorithm can distinguish the case that the label is pure noise, from the case that the label has a significant signal, using $o(\sqrt{d})$ datapoints (see e.g. Proposition 4.2 in [15]).

Our estimation machinery extends beyond the isotropic setting, and we prove an analog of Proposition 1 in the more general setting where the datapoints, $\mathbf{x}_i$ are drawn from a $d$ dimensional distribution with (unknown) non-isotropic covariance. This setting is considerably more challenging than the isotropic setting because the geometry of the datapoints depend heavily on the covariance, yet the covariance can not be accurately estimated with $o(d)$ samples. Though our results are weaker than in the isotropic setting, we still establish accurate estimation of the unexplained variance in the sublinear regime, though require a sample size $O_\epsilon(d^{1-\sqrt{\epsilon}})$ to obtain an estimate within error $O(\epsilon)$. In the case where the covariance matrix has a constant condition number, the sample size can be reduced to $n = O_\epsilon(d^{1-\frac{1}{\log 1/\epsilon}})$.

Our results in the non-isotropic setting apply to the following standard model of non-isotropic distributions: the distribution is specified by an arbitrary $d \times d$ real-valued matrix, $S$, and a univariate random variable $Z$ with mean 0, variance 1, and bounded fourth moment. Each sample $\mathbf{x} \in \mathbf{R}^d$ is then obtained by computing $\mathbf{x} = S\mathbf{z}$ where $\mathbf{z} \in \mathbf{R}^d$ has entries drawn independently according to $Z$. In this model, the covariance of $\mathbf{x}$ will be $SS^T$. This model is fairly general (by taking $Z$ to be a standard Gaussian this model can represent any $d$-dimensional Gaussian distribution, and it can also represent any rotated and scaled hypercube, etc), and is widely considered in the statistics literature (see e.g. [17, 1]). While our theoretical results rely on this modeling assumption, our algorithm is not tailored to this specific model, and likely performs well in more general settings.

**Theorem 1.** *Suppose we are given $n < d$ labeled examples, $(\mathbf{x}_1, y_1), \ldots, (\mathbf{x}_n, y_n)$, with $\mathbf{x}_i = S\mathbf{z}_i$ where $S$ is an unknown arbitrary $d \times d$ real matrix and each entry of $\mathbf{z}_i$ is drawn independently from a one dimensional distribution with mean zero, variance 1, and constant fourth moment. Assuming that each label $y_i = \mathbf{x}_i\beta + \eta$, where the noise $\eta$ is drawn independently from an unknown distribution with mean 0 and variance $\delta^2$, and the labels have been normalized to have unit variance. There is an algorithm that takes $n$ labeled samples, parameter $k$, $\sigma_{max}, \sigma_{min}$ which satisfies $\sigma_{max}I \succeq S^TS \succeq \sigma_{min}I$, and with probability $1 - \tau$, outputs an estimate $\hat{\delta}^2$ with additive error $|\hat{\delta}^2 - \delta^2| \leq \min(\frac{1}{k^2}, 2e^{-(k-1)\sqrt{\frac{\sigma_{min}}{\sigma_{max}}}})\sigma_{max}\|\beta\|^2 + \frac{f(k)}{\tau}\sum_{i=2}^k \frac{d^{i/2-1/2}}{n^{i/2}}$, where $f(k) = k^{O(k)}$.*

Considering the case when $\|\beta\|, \sigma_{max}$ are constants, setting $k = 1/\sqrt{\epsilon}$ in the case where $\sigma_{\min} = 0$ or $k = O(\log(\frac{1}{\epsilon}))$ in the case where $\sigma_{\min}$ is a constant greater than 0 yields the following corollary:

**Corollary 1.** *In the setting of Theorem 1, with constant $\|\beta\|$ and $\sigma_{max}$, the noise can be approximated to error $O(\epsilon)$ with $n = O(poly(\epsilon)d^{1-\sqrt{\epsilon}})$. With the additional assumption that $\sigma_{min}$ is a constant greater than $0$, the noise can be approximated to error $O(\epsilon)$ with $n = O(poly(\log(1/\epsilon))d^{1-\frac{1}{\log 1/\epsilon}})$.*

Finally, we establish the lower bound, demonstrating that, without any assumptions on $\|\Sigma\|$ or $\|\beta\|$, no sublinear sample estimation is possible. See Theorem 3 of the supplementary material for the lowerbound statement.

**The Agnostic Setting.** Our algorithms and techniques do not rely on the assumption that the labels consist of a linear function plus independent noise, and our results partially extend to the agnostic setting. Formally, assuming that the label, $y$, can have *any* joint distribution with $\mathbf{x}$, we show that our algorithms will accurately estimate the fraction of the variance in $y$ that can be explained via (the best) linear function of $\mathbf{x}$, namely the quantity $\inf_\beta \mathbf{E}[(\beta^T \mathbf{x} - y)^2]$. The analog of Proposition 1 in the agnostic setting is the following:

**Theorem 2.** *Suppose we are given $n$ labeled examples, $(\mathbf{x}_1, y_1), \ldots, (\mathbf{x}_n, y_n)$, with $(\mathbf{x}_i, y_i)$ drawn independently from a $d + 1$-dimension distribution where $\mathbf{x}_i$ has mean zero and identity covariance, and $y_i$ has mean zero and variance $1$, and the fourth moments of the joint distribution $(\mathbf{x}, y)$ is bounded by $C$. There is an estimator $\hat{\delta}^2$, that with probability $1 - \tau$, approximates $\inf_\beta \mathbf{E}[(\beta^T \mathbf{x} - y)^2]$ with additive error $O(C \frac{\sqrt{d+n}}{\tau n})$.*

In the setting where the distribution of $\mathbf{x}$ is non-isotropic, the algorithm to which Theorem 1 applies still extends to this agnostic setting. The estimate of the unexplained variance is still accurate in expectation, though additional assumptions on the (joint) distribution of $(\mathbf{x}, y)$ would be required to bound the variance of the estimator in this agnostic and non-isotropic setting. Such conditions are likely to be satisfied in many practical settings, though a fully general agnostic and non-isotropic analog of Theorem 1 likely does not hold.

**Binary Classification.** Our approaches and techniques for the linear regression setting also can be applied to the important setting of binary classification—namely estimating the performance of the best linear classifier, in the regime in which there is insufficient data to learn any accurate classifier. As an initial step along these lines, we obtain strong results in a restricted model of Gaussian data with labels corresponding to the latent variable interpretation of logistic regression:

**Theorem 3.** *Suppose we are given $n < d$ labeled examples, $(\mathbf{x}_1, y_1), \ldots, (\mathbf{x}_n, y_n)$, with $\mathbf{x}_i$ drawn independently from a Gaussian distribution with mean $0$ and covariance $\Sigma$ where $\Sigma$ is an unknown arbitrary $d$ by $d$ real matrix. Assuming that each label $y_i$ takes value $1$ with probability $g(\beta^T \mathbf{x}_i)$ and $-1$ with probability $1 - g(\beta^T \mathbf{x}_i)$, where $g(x) = \frac{1}{1+e^{-x}}$ is the sigmoid function and $\beta$ is an unknown parameter vector. There is an algorithm that takes $n$ labeled samples, parameter $k$, $\sigma_{max}$ and $\sigma_{min}$ which satisfies $\sigma_{max}I \succeq \Sigma \succeq \sigma_{min}I$, and with probability $1 - \tau$, outputs an estimate $\widehat{err_{opt}}$ with additive error $|\widehat{err_{opt}} - err_{opt}| \leq c\left(\sqrt{\min(\frac{1}{k^2}, 2e^{-(k-1)\sqrt{\frac{\sigma_{min}}{\sigma_{max}}}})\sigma_{max}\|\beta\|^2 + \frac{f(k)}{\tau}\sum_{i=2}^{k}\frac{d^{i/2-1/2}}{n^{i/2}}}\right)$, where $err_{opt}$ is the classification error of the best linear classifier, $f(k) = k^{O(k)}$ and $c$ is an absolute constant.*

In the setting where the distribution of $\mathbf{x}$ is an isotropic Gaussian, we obtain the simpler result that the classification error of the best linear classifier can be accurately estimated with $O(\sqrt{d})$ samples. This is information theoretically optimal, see Theorem 8 of the supplementary material.

**Corollary 2.** *In the setting of Theorem 3 but where each datapoint, $\mathbf{x}_i$ is drawn according to a $d$-dimension isotropic Gaussian distribution, there is an algorithm that takes $n$ labeled samples, and with probability $1 - \tau$, outputs an estimate $\widehat{err_{opt}}$ with additive error $|\widehat{err_{opt}} - err_{opt}| \leq c(\frac{\sqrt{d}}{n})^{1/2}$, where $err_{opt}$ is the classification error of the best linear classifier and $c$ is an absolute constant.*

Despite the strong assumptions on the data-generating distribution in the above theorem and corollary, the algorithm to which they apply seems to perform quite well on real-world data, and is capable of accurately estimating the classification error of the best linear predictor, even in the data regime where it is impossible to learn any good predictor. One partial explanation is that our approach can be easily adapted to a wide class of "link functions," beyond just the sigmoid function addressed by the above results. Additionally, for many smooth, monotonic functions, the resulting algorithm is almost identical to the algorithm corresponding to the sigmoid link function.

## 1.2 Related Work

For the specific question of estimating the signal to variance ratio (or signal to noise), also referred to as the "unexplained variance", there are many classic and more recent estimators that perform well in the linear and super-linear data regime. These estimators apply to the most restrictive setting we consider, where each label $y = \beta^T \mathbf{x} + \eta$ is given as a linear function of $\mathbf{x}$ plus independent noise $\eta$ of variance $\delta^2$. Two common estimators for $\delta^2$ involve first computing the parameter vector $\hat{\beta}$ that minimizes the squared error on the $n$ datapoints. These estimators are 1) the "naive estimator" or the "maximum likelihood" estimator: $(\mathbf{y} - X\hat{\beta})^T (\mathbf{y} - X\hat{\beta})/n$, and 2) the "unbiased" estimator $(\mathbf{y} - X\hat{\beta})^T (\mathbf{y} - X\hat{\beta})/(n - d)$, where $\mathbf{y}$ refers to the vector of $n$ labels, and $X$ is the $n \times d$ matrix whose rows represent the $n$ datapoints. Of course, both of these estimators are zero (or undefined) in the regime where $n \leq d$, as the prediction error $(\mathbf{y} - X\hat{\beta})$ is identically zero in this regime. Additionally, the variance of the unbiased estimator increases as $n$ approaches $d$, as is evident in our empirical experiments where we compare our estimators with this unbiased estimator. In the regime where $n < d$, variants of these estimators might still be applied but where $\hat{\beta}$ is computed as the solution to a regularized regression (see, e.g. [16]); however, such approaches seem unlikely to apply in the sublinear regime where $n = o(d)$, as the recovered parameter vector $\hat{\beta}$ is not significantly correlated with the true $\beta$ in this regime, unless strong assumptions are made on $\beta$.

Indeed, there has been a line of work on estimating the noise level $\delta^2$ assuming that $\beta$ is sparse [9, 7, 13, 12, 2]. These works give consistent estimates of $\delta^2$ in the regime where $n = \Omega(k \log(d))$ where $k$ is the sparsity of $\beta$. More generally, there is an enormous body of work on the related problem of *feature selection*.The basis dependent nature of this question (i.e. identifying which features are relevant) and the setting of sparse $\beta$, are quite different from the setting we consider where the signal may be a dense vector.

There have been recent results on estimating the variance of the noise, without assumptions on $\beta$, in the $n < d$ regime. In the case where $n < d$ but $n/d$ approaches a constant $c \leq 1$, Janson et al. proposed the EigenPrism [10] to estimate the noise level. Their theoretical results rely on the assumptions that the data $\mathbf{x}$ is drawn from an isotropic Gaussian distribution, and that the label is a linear function plus independent noise, and the performance bounds become trivial if $n/d \to 0$.

The most similar work to our paper is the work of Dicker [6], which proposed an estimator of $\delta^2$ with error rate $O(\frac{\sqrt{d}}{n})$ in the setting where the data $\mathbf{x}$ is drawn from an isotropic Gaussian distribution, and the label is a linear function plus independent Gaussian noise. Their estimator is fairly similar to ours in the identity covariances setting and gives the same error rate. However, our result is more general in the following senses: 1) Our estimator and analysis do not rely on Gaussianity assumptions; 2) Our results apply beyond the setting where the label $y$ is a linear function of $\mathbf{x}$ plus independent noise, and estimates the fraction of the variance that can be explained via a linear function (the "agnostic" setting); and 3) our approach extends to the unknown non-isotropic covariance setting.

In case the sparsity of $\beta$ is unknown, Verzelen and Gassiat [14] introduced a hybrid approach which combines Dicker's result in the dense regime and Lasso in the sparse regime to achieve consistent estimation of $\delta^2$ using $\min(k \log(d), \sqrt{d \log(d)})$ samples in the isotropic covariance setting where $k$ is the unknown sparsity of $\beta$, and they showed the optimality of the algorithm. In the unknown covariance, dense $\beta$ setting, they conjectured consistent estimation of $\delta^2$ is not possible with $o(d)$ samples; our Theorem 1 shows that this conjecture is false.

Finally, there is a body of work from the theoretical computer science community on "testing" whether a function belongs to a certain class, including work on testing *linearity* [3, 4] and *monotonicity* [8, 5]. Most of this work is in the "query model" where the algorithm can (adaptively) choose a point, $\mathbf{x}$, and obtain its label $\ell(\mathbf{x})$, and the flavor of results is very different from the setting we consider.

## 2 Intuition for Sublinear Estimation

We begin by describing one intuition for why it is possible to estimate the magnitude of the noise using only $O(\sqrt{d})$ samples, in the isotropic setting. Suppose we are given data $\mathbf{x}_1, \ldots, \mathbf{x}_n$ drawn i.i.d. from $N(0, I_d)$, and let $y_1, \ldots, y_n$ represent the labels, with $y_i = \beta^T \mathbf{x}_i + \eta$ for a random vector $\beta \in \mathbf{R}^d$ and $\eta$ drawn independently from $N(0, \delta^2)$. Fix $\beta$, and consider partitioning the datapoints

into two sets, according to whether the label is positive or negative. In the case where the labels are complete noise ($\delta^2 = 1$), the expected value of a positively labeled point is the same as that of a negatively labeled point and is $\overrightarrow{0}$. In the case where there is little noise, the expected value $\mu_+$ of a positive point will be different than that of a negative point, $\mu_-$, and the distance between these points corresponds to the distance between the mean of the 'top' half of a Gaussian and the 'bottom' half of a Gaussian. Furthermore, this distance between the expected means will smoothly vary between $0$ and $2\sqrt{2/\pi}$ as the variance of the noise, $\delta^2$, varies between 1 and 0.

The crux of the intuition for the ability to estimate $\delta^2$ in the regime where $n = O(\sqrt{d})$ is the following observation: while the empirical means of the positive and negative points have high variance in the $n = o(d)$ regime, it is possible to accurately estimate the *distance* between $\mu_+$ and $\mu_-$ from these empirical means! At a high level, this is because the empirical means consists of $d$ coordinates, each of which has a significant amount of noise. However, their squared distance is just a single number which is a sum of $d$ quantities, and we can leverage concentration in the amount of noise contributed by these $d$ summands to save a $\sqrt{d}$ factor. This closely mirrors the folklore result that it requires $O(d)$ samples to accurately estimate the mean of an identity covariance Gaussian with unknown mean, $N(\mu, I_d)$, though the norm of the mean $\|\mu\|$ can be estimated to error $\epsilon$ using only $n = O(\sqrt{d}/\epsilon)$.

Our actual estimators, even in the isotropic case, do not directly correspond to the intuitive argument sketched in this section. In particular, there is no partitioning of the data according to the sign of the label, and the unbiased estimator that we construct does not rely on any Gaussianity assumption.

## 3   The Estimators, Regression Setting

The basic idea of our proposed estimator is as follows. Given a joint distribution over $(\mathbf{x}, y)$ where $\mathbf{x}$ has mean 0 and variance $\Sigma$, the classical least squares estimator which minimizes the unexplained variance takes the form $\beta = \mathbf{E}[\mathbf{x}\mathbf{x}^T]^{-1}\mathbf{E}[y\mathbf{x}] = \Sigma^{-1}\mathbf{E}[y\mathbf{x}]$, and the corresponding value of the unexplained variance is $\mathbf{E}[(y - \beta^T\mathbf{x})^2] = \mathbf{E}[y^2] - \beta^T\Sigma\beta$. Note that this definition of $\beta$ coincides with the model parameter in the case that $y = \beta^T\mathbf{x} + \eta$. The variance of the labels, $y$ can be estimated up to $1/\sqrt{n}$ error with $n$ samples, after which the problem reduces to estimating $\beta^T\Sigma\beta$. While we do not have an unbiased estimator of $\beta^T\Sigma\beta$, as we show, we can construct an unbiased estimator for $\beta^T\Sigma^k\beta$ for any integer $k \geq 2$.

To see the utility of estimating these "higher moments", assume for simplicity that $\Sigma$ is a diagonal matrix. Consider the distribution over $\mathbf{R}$ consisting of $d$ point masses with the $i$th point mass located at $\Sigma_{i,i}$ with probability mass $\beta_i^2/\|\beta\|^2$. The problem of estimating $\beta^T\Sigma\beta$ is now precisely the problem of approximating the first moment of this distribution, and we are claiming that we can compute unbiased (and low variance) estimates of $\beta^T\Sigma^k\beta$ for $k = 2, 3, \ldots$, which exactly correspond to the 2nd, 3rd, etc. moments of this distribution of point masses. Our main theorem follows from the following two components: 1) There is an unbiased estimator that can estimate the $k$th ($k \geq 2$) moment of the distribution using only $O(d^{1-1/k})$ samples. 2) Given accurate estimates of the 2nd, 3rd,...,$k$th moments, one can approximate the first moment with error $O(1/k^2)$. The main technical challenge is the first component—constructing and analyzing the unbiased estimators for the higher moments. Analyzing the variance of these unbiased estimators is quite complicated, as a significant amount of machinery needs to be developed to deal with the combinatorial number of types of "cross-terms" in the variance expression for the estimators. Fortunately, we are able to leverage some techniques from [11], which bounds similar looking moments (with the rather different goal of recovering the covariance spectrum). The second component of our approach—leveraging estimates of the "higher moments" to estimate the "first moment" follows easily from standard results on polynomial approximation. Algorithm 1, to which Theorem 1 applies, describes our estimator in the general setting where the data has an arbitrary covariance matrix. The proof of correctness of Algorithm 1, establishing Theorem 1 is given in a self-contained form in the supplementary material.

In the special case where the data distribution has identity covariance, $\beta^T\Sigma^2\beta = \beta^T\Sigma\beta$ simply because $I^2 = I$, and hence we *do* have a simple unbiased estimator, which is constructed by Algorithm 1 by letting the input polynomial $p(x) = 1$. To see the intuition for the unbiased estimators of $\beta^T\Sigma^k\beta$ computed in Algorithm 1, we examine the $k = 2$ case: consider drawing two independent samples $(\mathbf{x}_1, y_1), (\mathbf{x}_2, y_2)$ from our linear model plus noise. Indeed, $y_1 y_2 \mathbf{x}_1^T \mathbf{x}_2$ is an unbiased estimator of $\beta^T\Sigma^2\beta$, because $\mathbf{E}[y_1 y_2 \mathbf{x}_1^T \mathbf{x}_2] = \mathbf{E}[y_1 \mathbf{x}_1^T]\mathbf{E}[y_2 \mathbf{x}_2] = \beta^T\Sigma^2\beta$. Given $n$ samples, by linearity of expectation, a natural unbiased estimate is hence to compute this quantity for each pair

---
**Algorithm 1** Estimating Linearity, General covariance
---

**Input**: $X = \begin{bmatrix} \mathbf{x}_1 \\ \vdots \\ \mathbf{x}_n \end{bmatrix}$, $\quad \mathbf{y} = \begin{bmatrix} y_1 \\ \vdots \\ y_n \end{bmatrix}$, and degree $k$ polynomial $p(x) = \sum_{i=0}^{k-2} a_i x^{i+2}$ that approxi-

mates the function $f(x) = x$ for all $x \in [\sigma_{\min}, \sigma_{\max}]$, where $\sigma_{\min}$ and $\sigma_{\max}$ are the minimum and maximum singular values of the covariance of the distribution from which the $\mathbf{x}_i$'s are drawn.

- Set $A = XX^T$, and let $G = A_{up}$ be the matrix A with the diagonal and lower triangular entries set to zero.

**Output**: $\frac{\mathbf{y}^T \mathbf{y}}{n} - \sum_{i=0}^{k-1} a_i \frac{\mathbf{y}^T G^{i+1} \mathbf{y}}{\binom{n}{i+2}}$

---

(of distinct) samples, and take the average of these $\binom{n}{2}$ quantities. This is precisely what Algorithm 1 computes, since $\mathbf{E}[\mathbf{y}^T G \mathbf{y}] = \mathbf{E}[\sum_{i<j} y_i y_j \mathbf{x}_i^T \mathbf{x}_j]$.

## 4 The Estimators, Binary Classification Setting

In the binary classification setting, we assume that we have $n$ independent labeled samples $(\mathbf{x}_1, y_1), \ldots, (\mathbf{x}_n, y_n)$ where each $\mathbf{x}_i$ is drawn from a Gaussian distribution $\mathbf{x}_i \sim N(0, \Sigma)$. There is an underlying link function $g : \mathcal{R} \to [0, 1]$ which is monotonically increasing and satisfies $g(0) = 1/2$, and an underlying weight vector $\beta$, such that each label $y_i$ takes value 1 with probability $g(\beta^T \mathbf{x}_i)$ and $-1$ with probability $1 - g(\beta^T \mathbf{x}_i)$. Under this assumption, the goal of our algorithm is to predict the classification error of the best linear classifier. In this setting, the best linear classifier is simply the linear threshold function $sgn(\beta^T \mathbf{x})$ whose classification error is $\frac{1}{2} - \mathbf{E}[|g(\beta^T \mathbf{x}) - \frac{1}{2}|] = \frac{1}{2} - \mathbf{E}_{x \sim N(0,1)}[|g(\|\beta^T \Sigma^{1/2}\|x) - \frac{1}{2}|]$.

The core of the estimators in the binary classification setting is the following observation: First, similar to the linear regression setting, we obtain a series of unbiased estimators for $\mathbf{E}[(g(\beta^T \mathbf{x}) - \frac{1}{2})\frac{\beta^T \mathbf{x}}{\beta^T \Sigma \beta}]^2 \beta^T \Sigma^k \beta$ with $k = 2, 3, \ldots$. Then, we find a linear combination of those estimates to yield an estimate of $\mathbf{E}[(g(\beta^T \mathbf{x}) - \frac{1}{2})\frac{\beta^T \mathbf{x}}{\beta^T \Sigma \beta}]^2 \beta^T \Sigma \beta = \mathbf{E}_{x \sim N(0,1)}[(g(\|\beta^T \Sigma^{1/2}\|x) - \frac{1}{2})x]^2$. Second the quantity we estimated in the first step $\mathbf{E}_{x \sim N(0,1)}[(g(\|\beta^T \Sigma^{1/2}\|x) - \frac{1}{2})x]^2$ is monotonically increasing in $\|\Sigma^{1/2}\beta\|$, and hence can be used to determine $\|\Sigma^{1/2}\beta\|$. The classification error, $\frac{1}{2} - \mathbf{E}_{x \sim N(0,1)}[|g(\|\Sigma^{1/2}\beta\|x) - \frac{1}{2}|]$, can then be calculated as a function of the estimate of $\|\Sigma^{1/2}\beta\|$. Our general covariance algorithm for estimating the classification error of the best linear predictor, to which Theorem 3 applies, is the following:

---
**Algorithm 2.** Estimating Classification Error, General Covariance
---

**Input**: $X = \begin{bmatrix} \mathbf{x}_1 \\ \vdots \\ \mathbf{x}_n \end{bmatrix}$, $\quad \mathbf{y} = \begin{bmatrix} y_1 \\ \vdots \\ y_n \end{bmatrix}$, degree $k-1$ polynomial $p(x) = \sum_{i=0}^{k-1} a_i x^i$ that approximates

the function $f(x) = x$ for all $x \in [\sigma_{\min}, \sigma_{\max}]$, where $\sigma_{\min}$ and $\sigma_{\max}$ are the minimum and maximum singular values of the covariance of the distribution from which the $\mathbf{x}_i$'s are drawn, and function $F_g$ which maps $\mathbf{E}_{x \sim N(0,1)}[(g(\|\beta^T \Sigma^{1/2}\|x) - \frac{1}{2})x]$ to $\frac{1}{2} - \mathbf{E}_{x \sim N(0,1)}[(g(\|\beta^T \Sigma^{1/2}\|x) - \frac{1}{2})]$. (Figure 1 in the supplementary material depicts $F_g$ in the case that $g$ is the sigmoid function.)

- Set $A = XX^T$, and let $G = A_{up}$ be the matrix A with the diagonal and lower triangular entries set to zero.

- Let $t = \frac{\sqrt{\sum_{i=0}^{k-1} a_i \frac{\mathbf{y}^T G^{i+1} \mathbf{y}}{\binom{n}{i+2}}}}{2}$, which is an estimate of $\mathbf{E}_{x \sim N(0,1)}[(g(\|\beta^T \Sigma^{1/2}\|x) - \frac{1}{2})x]$.

**Output**: $F_g(t)$

---

## 5 Empirical Results

We evaluated our estimators in both the regression and classification settings, on real and synthetic data. All experiments were run in Matlab v2016b on a MacBook Pro laptop, and the code is available from our websites. More details of the experiments are given in the supplementary material.

**Regression: Synthetic Data Experiments.** In this experiment, $n$ datapoints $\mathbf{x}_1, \ldots, \mathbf{x}_n \in \mathbf{R}^d$ are drawn from an multivariate Gaussian, $N(0, \Sigma)$. The labels $y_1, \ldots, y_n$ are computed by first selecting a uniformly random vector, $\beta$, scaling $\beta$ such $\mathbf{Var}[\beta^T x] = 1 - \delta^2$, and then setting each $y_i = \beta^T \mathbf{x}_i + \eta$ where $\eta$ is drawn independently from $N(0, \delta^2)$. The $y_i$'s are then scaled according to their empirical variance (simulating the setting where we do not know, a priori, that the labels have variance 1), and the magnitude of the fraction of this (unit) variance that is unexplained via a linear model is computed via Algorithm 1.

We evaluate Algorithm 1 in the isotropic covariance setting ($\Sigma = I$), depicted in Figure 1, and the non-isotropic covariance setting ($\Sigma$ with singular values $1/d, 2/d, 3/d, \ldots, 1$), depicted in Figure 2), each with three choices of the dimension, $d$=1,000, $d$=10,000, and $d$=50,000, and a range of choices of $n$ for each $d$. As expected, the results are more impressive for larger $n$, and demonstrate the ability of Algorithm 2 to perform well in the sublinear data setting where $n \ll d$.

**Regression: NLP Experiments.** We also evaluated our approach on an amusing natural language processing dataset: predicting the "point score" of a wine (the scale used by *Wine Spectator* to quantify the quality of a wine), based on a description of the tasting notes of the wine. This data is from Kaggle's Wine-Reviews dataset, originally scraped from Wine Spectator. The dataset contained data on 150,000 highly-rated wines, each of which had an integral point score in the range $[80, 100]$. The tasting notes consisted of several sentences, with a mean length of 40 words. The following is a typical tasting note (corresponding to a 96 point wine): *Ripe aromas of fig, blackberry and cassis are softened and sweetened by a slathering of oaky chocolate and vanilla. This is full, layered, intense and cushioned on the palate, with rich flavors of chocolaty black fruits and baking spices....*

Figure 3 explores the ability of a linear model to predict the point value of the wine, for two featurizations of the tasting note—corresponding to 1) concatenating the GloVe word vectors of the first 20 words in the tasting note, and 2) a 9k dimensional featurization corresponding to outerproducts of embeddings of pairs of words. This dataset is well-suited for our setting because the NLP setting presents a variety of natural high-dimensional featurizations, and the 150k datapoints were sufficient to accurately estimate a "ground truth" prediction error, allowing us to approximate the residual variance in the point value that cannot be captured via a linear model over the specified features.

**Binary Classification: Synthetic Data Experiments.** We evaluated Algorithm 2 on synthetic data with non-isotropic covariance. In this experiment, $n$ datapoints $\mathbf{x}_1, \ldots, \mathbf{x}_n \in \mathbf{R}^d$ are drawn from a uniformly randomly rotated Gaussian $G$ with covariance with singular values $1/d, 2/d, 3/d, \ldots, 1$. Model parameter $\beta$ is a $d$-dimensional vector with $\|\beta\| = 2$ that points in an uniformly random direction. Each label $y_1, \ldots, y_n$ is assigned by setting $y_i$ to be 1 with probability $g(\beta^T \mathbf{x}_i)$ and $-1$ with probability $1 - g(\beta^T \mathbf{x}_i)$, where $g(x)$ is the sigmoid function. We then applied Algorithm 3 with $k = 3$ moments. Figure 4 depicts the mean and standard deviation (over 50 trials) of the recovered estimates of the classification error of the best linear classifier. Again, the performance of our algorithm seem more impressive for larger $d$, and demonstrates the ability of Algorithm 2 to perform well in the sublinear sample setting where $n < d$ and the (regularized) logistic regression algorithm can not recover an accurate classifier.

**Binary Classification: MNIST.** We also evaluated Algorithm 2 for predicting the classification error on the MNIST dataset. We take digits "0"–"4" as positive examples and "5"–"9" as negative examples. Each image is represented as a $d = 784$ dimensional vector, and the data are 0 centered and scaled so the largest singular value of the sample covariance matrix is 1. As shown in the plot, even with $1,500 \approx 2d$ samples, the training error of the unregularized logistic regression is 0, meaning the data is perfectly separable, and the learned classifier does not generalize. Although the conditions of Theorem 3 obviously do not hold for the MNIST dataset, our algorithm still provides a reasonable estimate even with less than $400 \approx d/2$ samples.

**Acknowledgments:** This work was supported by NSF award CCF-1704417, an ONR Young Investigator Award, and a Sloan Research Fellowship.

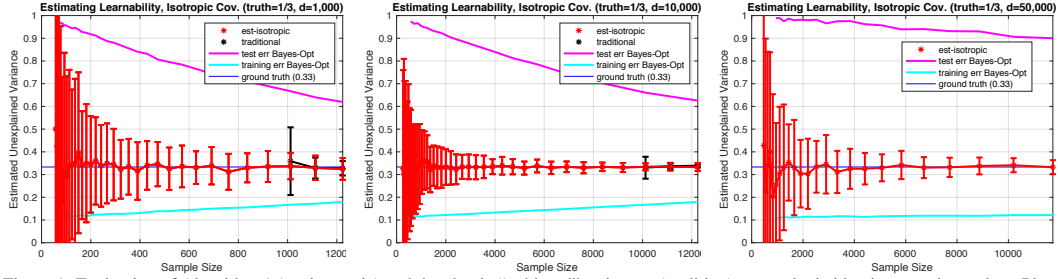

Figure 1: Evaluation of Algorithm 1 (est-isotropic) and the classic "unbiased" estimator (tradition) on synthetic identity-covariance data. Plots depict the mean and standard deviation (based on 50 trials) of the estimate of the fraction of the label variance that cannot be explained via a linear model, in a variety of parameter regimes. For comparison, we also included the test and training performance of the Bayes-optimal predictor (corresponding to $\ell_2$ regularized regression with optimal regularization parameter chosen as a function of the true variance of the noise).

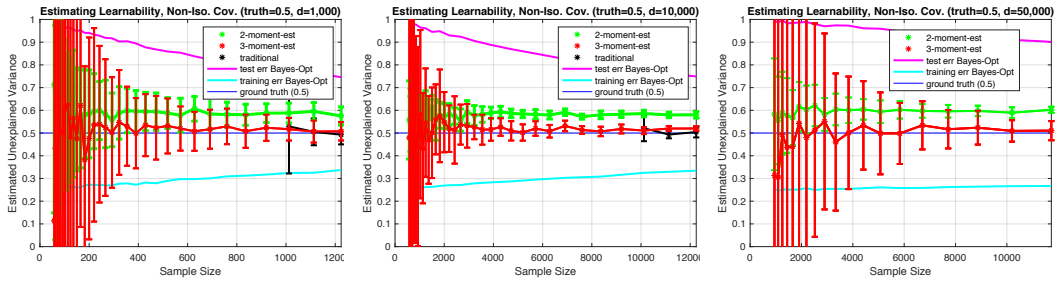

Figure 2: Evaluation of Algorithm 2 (using 2 and 3 moments) and the classic "unbiased" estimator (tradition) on synthetic data with covariance spectrum uniformly distributed between 0 and 1. Plots depict mean and standard deviation (based on 50 trials). As expected, the 2-moment estimator has a significant bias.The test and training performance of the Bayes-optimal predictor are also shown, for comparison.

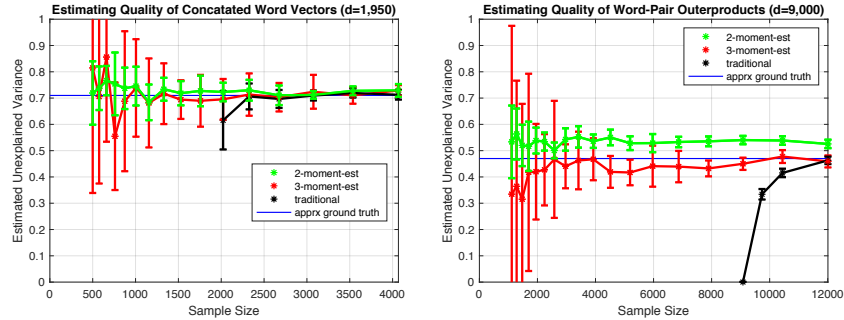

Figure 3: Evaluation of Algorithm 2 (using 2 and 3 moments) and the classic "unbiased" estimator (tradition) to predict the "point value' of a wine, based on a $\approx 40$ word "tasting note". Ground truth is estimated based on 150k datapoints. All data is from the Kaggle "Wine Reviews" dataset. The left plot depicts a naive featurization with $d = 1,950$, and the right plot depicts a quadratic embedding of pairs of words, with $d = 9,000$, which can explain more of the variance in the point scores.

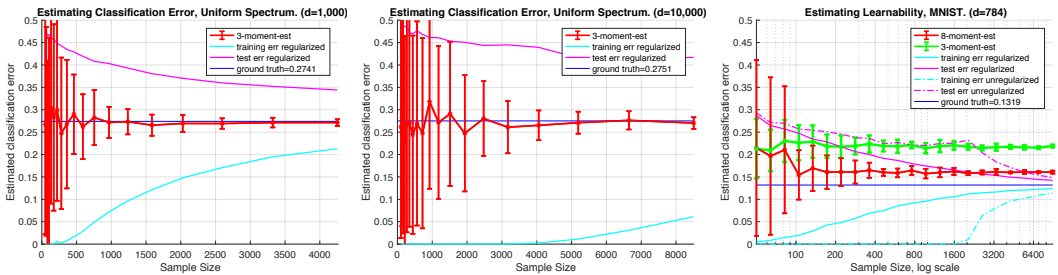

Figure 4: The left two plots depict the performance of Algorithm 2 using 3 moments (3-moment-est) and the $\ell_2$ regularized logistic regression estimator (training err regularized, test err regularized) on synthetic data with covariance spectrum uniformly distributed between 0 and 1. Plots depict mean and standard deviation (based on 50 trials). The rightmost plot depicts the performance of Algorithm 2 using 3 and 8 moments to predict the classification error of the best linear classifier on the MNIST dataset (to distinguish the class of digits $0 - 4$ versus $5 - 9$). For comparison, we plot the test and training error for unregularized logistic regression and $\ell_2$-regularized logistic regression. The ground truth is the average of the training and testing classification error based on training on 50k datapoints and testing on the remaining datapoints. Plots depict mean and standard deviation (based on 50 trials).

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
