[Reviews · NeurIPS 2018]

Reviewer 1



The authors pose and study an interesting question: Is it possible to use small amount of data to effectively estimate a likely performance metric of a predictor learned using large data? The authors show that the answer is yes, in some interesting cases. Assuming the data comes from an isotropic distribution in d-dimensions and the labels are arbitrary noisy functions of datapoints, they show that the variance of the best linear classifier can be accurately estimated using only O(sqrt(d)) samples. This is surprising because even in the noiseless setting, \Omega(d) sample size is required to learn any function coorelated with the underlying model. They show similar results for binary classification. The technical part and the idea of using higher moment estimation to approximate the first moment is very interesting. The authors also have empirical resuls to support their claims. Overall, I find this a clear accept.

Reviewer 2



The authors study the problem of estimating the noise in settings of linear regression. More precisely, given (x,y) pairs which are generated as y = beta*x + Z where Z ~ N(0,delta^2), how many samples are needed to estimate the value of delta? This is useful to estimate how many samples are required to truly perform accurate regression. Interestingly, when the x's are distributed as a Gaussian (say), sqrt(d) samples are sufficient to get an estimate. Note that even with no noise, d samples would be required to learn the classifier, as otherwise the system may be underdetermined. The authors also have results for arbitrary covariances, binary classification, matching lower bounds, and some neat experimental evaluation. This is pretty much a prototypical example of what I want to see in an ICML paper. The problem is natural and meaningful, the method is elegant and practical and requires some non-trivial technical insight, and the experiments are compelling and interesting. The paper is well written for the most part. The goal of the algorithm is to estimate the variance of the added noise. The method is to estimate the variance of the labels, and then subtract out the variance due to the variance in the data, leaving the variance of the noise. In general, it isn't clear how to design an unbiased estimator for the label variance, so instead the authors use higher order moments (which are estimatable) and carefully combine them to get an estimator for the label variance with low bias. This is a neat method, which has also been similarly used by Kong and Valiant. It's nice to see another application of this method, and this paper might be more illustrative than the other (as the problem is a bit less cumbersome, in my mind).

Reviewer 3



When I started reading the paper, I actually liked it. But when I got to page 5, I was very disappointed and annoyed. It might not be intentional, but it felt almost deceptive that the authors did not mention very relevant prior art till then and made it look like many of the previous literature as their own contribution. Authors claim that they propose an algorithm that uses O(sqrt(d)) samples for estimating the variance of noise. In fact, after a read of this paper and previous papers, what I got was: O(sqrt(d)) was essentially known before (for most cases) and new results are in fact much weaker. Dicker et al: Maximum Likelihood for Variance Estimation in High-Dimensional Linear Models (http://www.stat.rutgers.edu/home/ldicker/papers/variance.pdf), showed that O(sqrt(d)) samples are sufficient for learning sublinearity when additional noise (eta) is Gaussian i.i.d. and data points X are Gaussian and either have identity covariance or any general covariance (Lemma 2 and proposition 1 respectively). Further, for the convergence proof, the properties of Gaussianity they used were only that the moments are bounded. Hence their results and proofs more or less extend to non-Gaussian cases with similar moment bounds. What the authors showed: a. Extend the results of Dicker et al (with a very similar algorithm) to non-Gaussian but isotropic and bounded moments. But (mostly) the exact proof of Lemma 2 in Dicker et al holds for any isotropic distributions with bounded moments already. b. Theorem 1: They propose a weaker version of the result when the X is not from a Gaussian and is non-isotropic. Here the result is O(d^(1-g(epsilon)), which is much weaker than sqrt(d). Further the convergence rate is weak 1/poly(log(d/n)) as opposed to the polynomial convergence rates of Dicker et al. The dependence of epsilon is also bad and super linear. (exponential in 1/epsilon in certain cases). c. Extend isotropic results to the agnostic case, but here again, I am not sure what is the novelty compared to Lemma 2 in Dicker et al. Also authors did not compare Dicker et al algorithms with the “new” one in experiments. I request the authors to a. Exactly state what is new in Algorithm 1 compared to the algorithm of Dicker et al. b. Why are Theorem 1 results much weaker than the Proposition 1 results of Dicker et al.? c. What is the novelty in Proposition 1 and Theorem 2 compared to the proof of Lemma 2 in Dicker et al. d. State the convergence rate of the algorithm (Theorem 1). e. If the results are different than Dicker et al, include their algorithms in your experimental plots and compare experimentally which one works better.